# SCoMoE: Efficient Mixtures of Experts with Structured Communication

**Zhiyuan Zeng**[†]
Tianjin University
zhiyuan_zeng@tju.edu.cn

**Deyi Xiong**[*]
Tianjin University, GTCOM
dyxiong@tju.edu.cn

## Abstract

Mixture-of-Experts (MoE) models are promising architectures for massively multilingual neural machine translation and large language models due to the advantage of sublinear scaling. However, the training of large MoE models is usually bottlenecked by the all-to-all communication (Lepikhin et al., 2020). To reduce the communication cost, we propose SCoMoE, an MoE architecture with structured all-to-all communication, inspired by the hierarchical architecture of the communication topology. SCoMoE encourages data to be communicated across devices through fast intra-accelerator/node communication channels, reducing communication throughput in the slow inter-node communication channel. We slice the data on the sequence dimension (SCoMoE-Seq) into three communication groups and project the data on the feature dimension (SCoMoE-Feat) into low-dimensional representations. To compensate the potential performance drop caused by the routing locality in SCoMoE, we further propose a token clustering approach to aggregating related tokens from different devices before the MoE layers. The sigmoid gating in the balanced router used in the token clustering is substituted with the softmax gating with differential sorting. Experiments on bilingual and massively multilingual machine translation demonstrate that SCoMoE achieves a speedup of 1.44x over GShard with comparable performance, and substantially outperforms Gshard (2.8 BLEU) on OPUS-100 with a speedup of 1.25x. Codes are available at `https://github.com/ZhiYuanZeng/fairseq-moe`.

## 1 Introduction

Recent years have witnessed a substantial interest in exploring sparse architectures based on Mixture of Experts for training massively multilingual machine translation (Lepikhin et al., 2020; Kim et al., 2021) and large language models (Fedus et al., 2021; Zhang et al., 2021b; Ma et al., 2022; Du et al., 2021; Zoph et al., 2022; Rajbhandari et al., 2022; Lin et al., 2021). Experts of MoE models are distributed over multiple devices. Due to the sparse architecture where only a combination of experts are selected to process each input, the number of experts and hence the scale of MoE models can be sufficiently large while the computational cost is only sublinear to the number of parameters. Despite the advantage of efficient computation, MoE models require expensive all-to-all communication, to send the inputs and outputs of experts across the compute network. Previous study on GShard (Lepikhin et al., 2020) has shown that as MoE models scale, the all-to-all communication cost becomes the bottleneck for training.

To mitigate this issue, we propose Structured Communication based MoE (ScoMoE), which treats the all-to-all communication in a structured way rather than equally across different devices. The motivation behind SCoMoE is that the network bandwidth between devices and nodes is different across the compute network: the bandwidth inside an accelerator (intra-accelerator) is faster than that across accelerators, and the bandwidth inside a node (intra-node) is faster than that across nodes (inter-node). Figure 1a visualizes the hierarchical structure of communication topology with a $9 \times 9$ matrix, where different levels of communication are in different colors. We view the data flow in

---

[†]Work was done while the author was interning at GTCOM.
[*]Corresponding author.

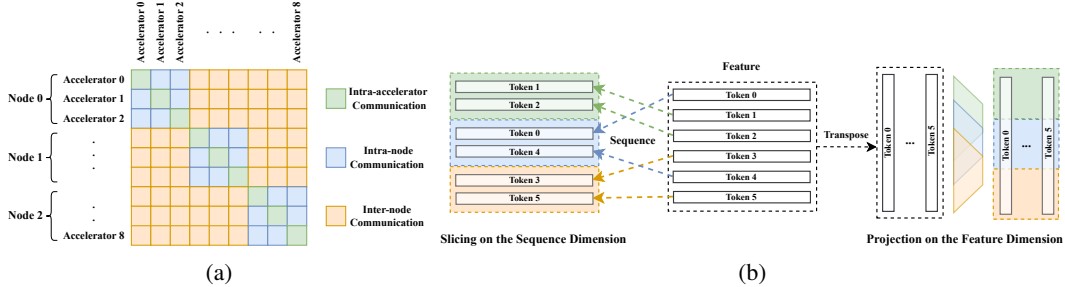

Figure 1: (a): The all-to-all communication contains three levels of communication with different bandwidth: the fast intra-accelerator communication inside each accelerator (green squares), the intra-node communication between Accelerator 0, 1 and Accelerator 3, 4 (blue squares), and the slow inter-node communication between Node 0 and Node 1 (orange squares). (b): Slicing data on the sequence dimension (left) and Projecting data on the feature dimension (right) into three groups corresponding to three levels of communication. Each row of the data is a token embedding.

the all-to-all communication from the perspective of two dimensions: sequence dimension (tokens) and feature dimension (embeddings of tokens). The proposed SCoMoE transforms (slicing or projecting) the data flow at either the sequence or feature dimension into three communication groups: intra-accelerator, intra-node and global (inter-node) communication, as shown in Figure 1b. For the data slicing on the sequence dimension, we select tokens for a communication group according to the assignment scores of the tokens with the experts inside the group. To organize the data transformation on the feature dimension, we linearly project features into the three communication groups with lower feature dimensionality and recast them back after the all-to-all communication.

Theoretically, structuring all-to-all communication in this way is faster than its original form, since less data are transmitted through the slow inter-node communication channel. However, this may hurt the performance, because intra-accelerator/intra-node communication can be processed by only a part of experts. To alleviate this issue, we further propose a token clustering approach to aggregating related tokens from different devices, which elevates the association of tokens inside each device (accelerator/node). The proposed token clustering uses the balance router presented by Lewis et al. (2021) for clustering, where each device is a cluster. In the balanced router, a sigmoid gate is adopted to combine the inputs and outputs of experts, which could only broadcast the gradients to the activated experts, even though it may be more suitable to dispatch the tokens to other experts. Hence we propose to replace the sigmoid gate with the softmax gate via a straight-through trick (Bengio et al., 2013) for better gradient broadcasting.

In a nutshell, our contributions are summarized as follows:

1. We propose SCoMoE that transforms the data flow in the all-to-all communication into three groups according to the bandwidth structure of communication topology.

2. A token clustering method is proposed to dispatch the related tokens to the same devices to alleviate the routing locality in the structured communication.

3. We propose to use softmax gate to substitue the sigmoid gate in the balanced router for better gradient broadcasting.

4. Experiments on bilingual and massively multilingual machine translation demonstrate that SCoMoE is faster than Gshard (Lepikhin et al., 2020) significantly with comparable or even better translation performance. Further analysis discloses the strategies of selecting hyper-parameters for SCoMoE.

## 2 RELATED WORK

MoE models (Jacobs et al., 1991; Jordan & Jacobs, 1994) are ensemble methods to integrate multiple experts. Shazeer et al. (2017) propose a gating network to select a combination of experts and mix data parallelism and model parallelism to increase the batch size. Gshard (Lepikhin et al., 2020) utilizes the MoE parallelism proposed by Shazeer et al. (2017) to scale Transformer by replacing

the MLP layer of Transformer with multiple MLP experts. Switch Transformer (Fedus et al., 2021) substitutes the top-2 routing of Gshard with the top-1 routing, and trains large sparse models with a selective precision strategy. Deepspeed-MoE (Kim et al., 2021) combines ZerO (Rajbhandari et al., 2020) and MoE parallelism to train large MoE models. Many recent large language models (Fedus et al., 2021; Zhang et al., 2021b; Ma et al., 2022; Du et al., 2021; Zoph et al., 2022) have been trained with Gshard and its variants. A variety of new routers, e.g., hash router (Roller et al., 2021), random router (Zuo et al., 2021) and balanced router (Lewis et al., 2021), have been proposed in lieu of the top-2 router in GShard. We employ the balanced router proposed by the BASE Layer (Lewis et al., 2021) for token clustering. The balanced router uses the auction algorithm (Bertsekas, 1992) for balanced routing.

Although sparse MoEs are computationally efficient, they suffer from frequent and expensive all-to-all communication across the network. Gshard (Lepikhin et al., 2020) has shown that the all-to-all communication becomes the bottleneck for training large MoEs. While the heterogeneous hierarchical network topology has been exploited to reduce the communication consumption, previous studies (Lin et al., 2020; Castiglia et al., 2021) focus on the all-reduce communication in SGD rather than the all-to-all communication in MoE. Tutel (Hwang et al., 2022) proposes a hierarchical all-to-all communication approach for MoE, which performs all-to-all communication and layout transformation inside a node, and then all-to-all communication across nodes. Although the hierarchical all-to-all communication of Tutel reduces the number of communication hops, it increases the amount of communication data. Hetu-MoE (Nie et al., 2022) introduces a hierarchical all-to-all communication mechanism similar to that of Tutel. The difference is that Hetu-MoE dispatches all data inside one node to one accelerator, which may cause memory overflow. Sparse-MLP (Lou et al., 2022) shortens the communication time by eliminating 10% tokens before routing, according to the importance scores between tokens and experts. However, removing tokens may hurt the performance. Fast-MoE (He et al., 2021) optimizes the training speed of MoE on the computation of experts, rather than communication. Task-level routing (Zhou et al., 2022) extracts a subset of experts for inference which improves the inference efficiency, but only works for multilingual/multi-task MoEs and the extracted submodel can only translate in one direction. Expert Choice Routing (Kudugunta et al., 2021) speedups the convergence speed of MoE with better load balance, but cannot reduce the time cost for each step.

**Comparison with Tutel**  Although both SCoMoE and Tutel improve the all-to-all communication of MoE based on the hierarchical network topology, SCoMoE is significantly different from Tutel in the following four aspects: 1) Motivation. Tutel attempts to decrease the communication hops and increase the message size for inter-node communication while SCoMoE is proposed to decrease the amount of data for inter-node communication and increase that for intra-node communication. 2) Algorithm. The communication of Tutel is hierarchical, first intra-node, and then inter-node. However, the communication of SCoMoE is structured, where different data are communicated in different groups (intra-accelerator/intra-node/inter-node). 3) Translation Performance. The hierarchical all-to-all communication algorithm proposed by Tutel may improve the communication speed, but does not influence the translation performance. By contrast, SCoMoE changes the routing of some data, which may serve as regularization and improve the translation performance. 4) Speedup. In certain situations, Tutel even slows down training while SCoMoE still improves training speed. The comparison of the two methods on the training speed are shown in Figure 3d. Despite these significant differences, SCoMoE is orthogonal to Tutel and the two methods can be easily combined.

**Massively Multilingual NMT**  Our work is also related to massively multilingual neural machine translation, which is briefly reviewed in Appendix A due to the constraint of space.

## 3  MODEL

### 3.1  SPARSE MIXTURE-OF-EXPERTS AND ALL-TO-ALL COMMINICATION

Sparse MoE models usually extend Transformer architecture with MoE layers where only a combination of experts are activated for computation each time. Different experts do not share parameters, as shown in Figure 2 (in blue), while other modules (in red) are shared across all devices.

Given an input sequence $\boldsymbol{X}$ which contains $n$ tokens $\{\boldsymbol{X}_1, ..., \boldsymbol{X}_i, ..., \boldsymbol{X}_n\}$ with $n$ token embeddings, sparse MoE models assign each token $\boldsymbol{X}_i$ to experts usually according to a token-to-expert assignment matrix $\boldsymbol{M}$:

$$\boldsymbol{M} = \boldsymbol{X}\boldsymbol{W_E}$$
$$\boldsymbol{M}_{ij} = {\boldsymbol{W_{E_j}}}^{\top} \cdot \boldsymbol{X}_i \tag{1}$$

$\boldsymbol{W_E}$ is a projection matrix, where each column is corresponding to an expert embedding. $\boldsymbol{M}_{ij}$ is the assignment score between an expert embedding ${\boldsymbol{W_{E_j}}}^{\top}$ and a token embedding $\boldsymbol{X}_i$. Given the token-to-expert assignment matrix $\boldsymbol{M}$, the top-2 gating router selects the top-2 experts for each input token, while the balanced router assigns tokens to experts with the auction algorithm.

After the assignment, tokens are sent to their target experts via the all-to-all communication across devices, where each device sends/receives (i.e., dispatches and combines) tokens to/from all other devices. Experts in devices process the received tokens, and send the processed tokens back to their original devices, which requires another all-to-all communication. The all-to-all communication contains three levels of communication: intra-accelerator, intra-node, and inter-node, visualized in green, blue and orange respectively in Figure 1a. The intra-accelerator communication is actually copying data from the memory inside the corresponding accelerator, which is very fast. The intra-node communication is the communication across different accelerators inside one node, which is much faster than the inter-node communication across different nodes. However, most of the communication in the all-to-all communication are the slowest inter-node communication, which is visualized in orange in Figure 1a. Furthermore, the ratio of inter-node communication increases quadratically with the number of nodes. Therefore, the all-to-all communication becomes very expensive and hence the bottleneck for training large MoE models with many devices.

## 3.2 STRUCTURED ALL-TO-ALL COMMUNICATION

To reduce the all-to-all communication cost, we propose to transform the tokens for cross-device communication into three groups corresponding to the three levels of communication: intra-accelerator, intra-node and global communication.[1] The key idea is to increase the amount of tokens dispatched/combined in the fast intra-accelerator/node communication channels, and decrease the amount of tokens in the slow inter-node communication channel. Intuitively, this structured all-to-all communication acts like the multi-head attention where the inputs are projected into multiple keys, queries and values before attention computation in that our structured communication transforms the data into multiple groups for communication at different levels before expert computation.

Different from the vanilla MoE, we do not compute the token-to-expert assignment scores for all experts. Instead, we aim to select the experts inside each accelerator for the intra-accelerator communication, and the experts inside each node for the intra-node communication. As shown in Figure 2, the tokens in the intra-accelerator/node group can only be processed by the experts inside the accelerator/node, while the tokens in the global group are processed by all experts. The processed tokens are sent back to their original devices.

## 3.3 DATA TRANSFORMATION FOR STRUCTURED COMMUNICATION

The data transformation for the structured communication could be either on the sequence or feature dimension. The difference of them is visualized in Figure 1b.

**Data Slicing on the Sequence Dimension** For the data transformation on the sequence dimension, we slice the tokens into each communication group according to the token-to-expert assignment scores introduced in the Section 3.1. The tokens to be sent to intra-accelerator/node devices have the highest assignment score with the experts inside the devices. Specifically, the tokens for local devices are collected via:

$$\mathbb{X}_{\text{local}} = \arg \operatorname*{TopK}_{i} \sum_{j \in \mathbb{E}_{\text{local}}} \boldsymbol{M}_{ij} \tag{2}$$

where $\boldsymbol{M}_{ij}$ is the assignment score between expert $j$ and token $i$, $\mathbb{E}_{\text{local}}$ is a set of experts distributed at local devices, $\mathbb{X}_{\text{local}}$ is a set of tokens that have the top-$k$ assignment scores with $\mathbb{E}_{\text{local}}$. $k$ denotes

---

[1]We do not perform data transformation for inter-node communication.

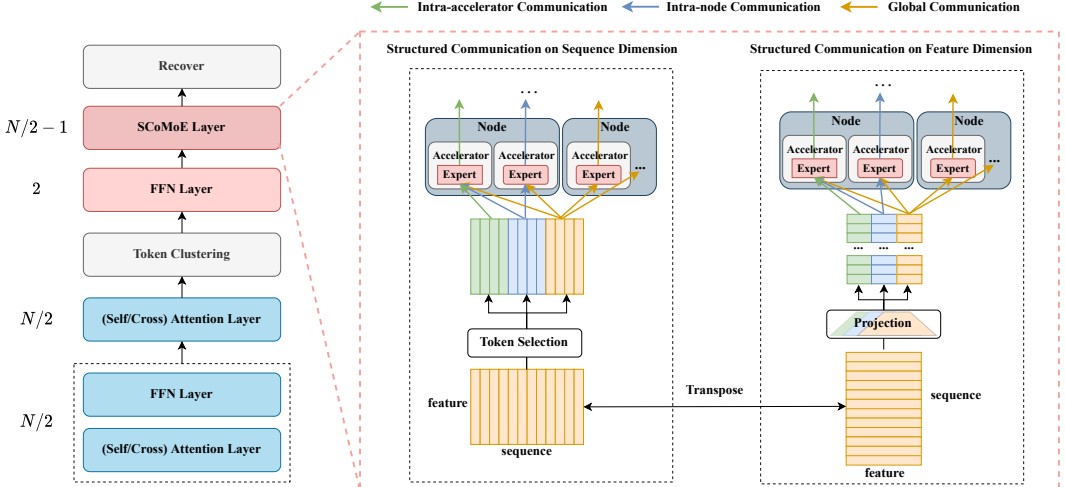

Figure 2: Architecture of the SCoMoE. Left: the overall structure of the model. Right: the detailed structure of a SCoMoE layer. There are $2N$ layers in the model, $N$ attention layers, $\frac{N}{2}$ shared FNN layers and $\frac{N}{2}$ MoE layers (token clustering with the 2 succeeding FFN layers can be seen as one MoE layer). The parameters of layers in blue are shared across different devices, while the parameters of layers in red are expert parameters. The token clustering aggregates related tokens at different devices together before structured communication layers. The aggregated tokens are sent back to the original devices (Recover) after the SCoMoE layers. The SCoMoE layers transform the data into three groups on either the sequence or feature dimension, and dispatch the transformed data to the experts via the intra-accelerator, intra-node and global communication respectively. We omit the process of combining the processed tokens into their original devices in the diagram for simplicity.

the number of tokens to be dispatched to local devices, which is a hyper-parameter. The tokens for both the intra-accelerator and intra-node communication are collected in this way. We primarily select token for local communication, first intra-accelerator, then intra-node and finally for global communication. We mask the token-to-expert assignment scores of the selected tokens with $-\infty$ to prevent the same tokens from being selected for multiple groups of communication.

**Data Projection on the Feature Dimension**   To transform the data on the feature dimension, we project the input sequence $\boldsymbol{X}$ with the dimensionality of $d_{\text{model}}$ to lower-dimensional representations of $d_{\text{intra-accelerator}}$, $d_{\text{intra-node}}$ and $d_{\text{global}}$, with three different projections $\boldsymbol{W}_{\text{intra-accelerator}}$, $\boldsymbol{W}_{\text{intra-node}}$ and $\boldsymbol{W}_{\text{global}}$:

$$
\begin{aligned}
\boldsymbol{X}_{\text{intra-accelerator}} &= \boldsymbol{X}\boldsymbol{W}_{\text{intra-accelerator}} \\
\boldsymbol{X}_{\text{intra-node}} &= \boldsymbol{X}\boldsymbol{W}_{\text{intra-node}} \\
\boldsymbol{X}_{\text{global}} &= \boldsymbol{X}\boldsymbol{W}_{\text{global}} \\
d_{\text{intra-accelerator}} + d_{\text{intra-node}} &+ d_{\text{global}} = d_{\text{model}}
\end{aligned}
\tag{3}
$$

where $\boldsymbol{X} \in \mathbb{R}^{n \times d_{\text{model}}}$, $\boldsymbol{X}_{\text{intra-accelerator}} \in \mathbb{R}^{n \times d_{\text{intra-accelerator}}}$, $\boldsymbol{X}_{\text{intra-node}} \in \mathbb{R}^{n \times n_{\text{intra-node}}}$, $\boldsymbol{X}_{\text{global}} \in \mathbb{R}^{n \times d_{\text{global}}}$. $\boldsymbol{X}_{\text{intra-accelerator}}$, $\boldsymbol{X}_{\text{intra-node}}$, $\boldsymbol{X}_{\text{global}}$ are projected sequences to be dispatched via different communication channels. We display these projections and projected sequences in different colors in Figure 2.

The dimensionality of data dispatched to experts is reduced into $d_{\text{intra-accelerator}}$, $d_{\text{intra-node}}$ or $d_{\text{global}}$. Hence we also need to reduce the dimensionality of weights in experts correspondingly. As an expert is actually a fully-connected network, which contains two linear projections $\boldsymbol{W}_1 \in \mathbb{R}^{d_{\text{model}} \times d_{\text{ffn}}}, \boldsymbol{W}_2 \in \mathbb{R}^{d_{\text{ffn}} \times d_{\text{model}}}$.[2] Specifically, we set the shape of the first projection to be $(\hat{d}, d_{\text{ffn}})$, and that of the second projection to be $(d_{\text{ffn}}, \hat{d})$. $\hat{d} \in \{d_{\text{intra-accelerator}}, d_{\text{intra-node}}, d_{\text{global}}\}$, corresponding

---

[2]We ignore the bias for simplicity.

to the input dimension. Actually, the parameters of the experts are projected into three parts, each of which is corresponding to one level of communication. The outputs of experts are recast back to the original dimension ($d_{\text{model}}$) with another three projections.

### 3.4 TOKEN CLUSTERING

The data slicing on the sequence dimension forces more tokens to be dispatched to local devices, which may hurt the performance, as tokens dispatched to local devices are processed by only a few experts inside the devices. To mitigate this issue, we propose to apply token clustering before the SCoMoE layers. The token clustering is to aggregate related tokens to the same device (accelerator/node), where each device is a cluster. Aggregating related tokens together will make it better for the experts to process them. But the router of MoE usually assigns related tokens to the experts (Zoph et al., 2022), why do we need the token clustering? The reason is that the router of SCo-MoE assigns more tokens to the experts inside local devices, while token clustering can mitigate this problem. It gathers the tokens that are related to the experts at the same devices together so that the tokens at the same devices may have a stronger association with each other and can be better processed by the same experts. We perform the token clustering before the first SCoMoE layer, and recover the original token distribution after the last SCoMoE layer.

We average the embeddings of experts inside an accelerator/node as the embedding of the device (either the accelerator or node), and assign tokens to devices in the same way as we assign tokens to experts. But unfortunately, the commonly used top-2 gating (Lepikhin et al., 2020) and top-1 gating (Fedus et al., 2021) routers could not be used here, since these routers abandon some tokens during routing. Therefore we resort to routers that keep all tokens, e.g, greedy router (Lewis et al., 2021) and balanced router (Lewis et al., 2021). The greedy router allows tokens to be dispatched to the top-1 expert according to the token-to-expert assignment matrix. However, different from the commonly used top-1 gating (Fedus et al., 2021), the greedy router does not set capacity for each expert, allowing any number of tokens to be sent to the same expert. When the greedy router is used to route tokens to devices, it may send too many tokens to a single device as it is unbalanced, causing out-of-memory issue. Incorporating the load balance loss (Lepikhin et al., 2020) does not help in this case, since the out-of-memory issue may immediately happen at the beginning of training. In contrast, the balanced router does not have the problem of the greedy router, which uses the auction algorithm (Bertsekas, 1992) to assign the same number of tokens to each expert. But the auction algorithm usually converges until many iterations, which is inefficient for autoregressive decoding. Hence the BASE Layer (Lewis et al., 2021) uses the balanced router instead of the greedy router at inference. In this work, we perform token clustering with the balanced router. However, we have found a weakness of Base Layer in gradient broadcasting, and proposed **Differentiable Sorting** to address it, which is described in Appendix B.

The token clustering aggregates the tokens of different sentences together, which cannot be input to the self-attention layer. Therefore we have to change the order of Transformer layers, putting the attention layers before the MoE layers. The new architecture is shown in Figure 2, where the shared encoder/decoder layers are placed at the bottom, followed by the attention layers and MoE layers. The design of the new architecture is due to the following reasons: (1) Press et al. (2020) have investigated the impact of the order of attention layers and feed-forward layers on the performance and found that putting the attention layers before the feed-forward layers often yields good performance. (2) Riquelme et al. (2021) find that the top MoE layers are more important than the bottom MoE layers. (3) Intuitively, the training of sparse MoE layers is more unstable than the training of the dense layers (Zoph et al., 2022). Hence, putting the stable dense layers at the bottom may help stabilize the training of the entire architecture.

We put the balanced router before the first SCoMoE layer for token clustering. However, the token clustering also requires the all-to-all communication. We remove one MoE layer to avoid extra communication cost. Hence the number of SCoMoE layers is $N/2 - 1$ rather than $N/2$, as shown in Figure 2. To keep the same computation cost with the original model, we also insert two fully-connected layers after the balanced router.[3] The balanced router together with the two fully-connected layers

---

[3]Two fully-connected layer rather than one, since we use the top-2 gating in SCoMoE layers, where each token is processed by two experts.

act as the first MoE layer. Since the token clustering works on the sequence dimension, we only apply the token clustering to the SCoMoE layer that slices the data on the sequence dimension.

## 4 EXPERIMENTS

### 4.1 EXPERIMENT SETTINGS

We evaluated the proposed SCoMoE on both bilingual and massively multilingual neural machine translation. In bilingual NMT experiments, we used the WMT17-En-Fr corpus, which consists of 35,762,532 training sentences, while in the massively multilingual NMT experiments, the OPUS-100 corpus (Zhang et al., 2020) was used, which consists of 100 languages and 107,924,846 training examples. The details of these two datasets are shown in Table 2 of Appendix C. We used the script of fairseq [4] to download and pre-process the WMT17-En-Fr dataset while the OPUS-100 dataset was directly downloaded from the OPUS web.[5] Both the WMT17-En-Fr and OPUS-100 dataset were tokenized via the BPE algorithm (Sennrich et al., 2016). For the multilingual experiments, a language token indicating the source language was prepended to each source sentence and similarly, a target language token was prepended to each target sentence.

Models were trained in two settings: base model and large model. The base models have 6 encoder/decoder layers, 8 experts and $d_{\text{model}} = 512$, which were trained on one node with 8 RTX A6000 GPUs. The large models have 12 encoder/decoder layers, 32 experts, and $d_{\text{model}} = 1024$, which were trained on 4 nodes with 32 RTX A6000 GPUs. The 4 nodes were connected with Infiniband, the bandwidth of which is 100 Gbps. The number of parameters in the base model is 0.2 billion while 4.1 billion in the large model. The details of the base and large model configuration could be found in Table 3 of Appendix C.

We compared our method with Gshard and its variant with a small capacity factor on translation performance and training speed. We refer to our SCoMoE on the sequence dimension as SCoMoE-Seq, and that on the feature dimension as SCoMoE-Feat. The Gshard with a small capacity factor is referred to as Small-Cap-MoE. We set the ratio of tokens dispatched through the intra-accelerator communication to be $\alpha$ ($0 < \alpha < 1$), and the ratio through the intra-node communication to be $\beta$ ($0 < \beta < 1$), hence that through the global communication is $\gamma = 1 - \alpha - \beta$ ($0 < \gamma < 1$). To simplify the comparison, we set either $\alpha$ or $\beta$ to be non-zero, and the other to be 0. We refer the SCoMoE-(Seq/Feat) with $\alpha > 0, \beta = 0, \gamma > 0$ as SCoMoE-(Seq/Feat)-$\alpha$, while that with $\alpha = 0, \beta > 0, \gamma > 0$ as SCoMoE-(Seq/Feat)-$\beta$. Please refer to Appendix C for more details about experiment settings.

### 4.2 MAIN RESULTS

We evaluated the proposed methods on the WMT17-En-Fr and OPUS-100. We report the BLEU score on the WMT17-En-Fr, and the average BLEU scores over all, English-to-any, and any-to-English language pairs on the OPUS-100. Results are shown in Table 1. We also report the win-rate and the speedup of all models over Gshard.

We evaluated the training speed on the dummy-mt task [6] of fairseq, since the training speed evaluated on it is more stable. Each input sentence in the dummy-mt task is a series of numbers, e.g., "0, 1, 2, 3, 4, ...", rather than the real sentence. We fixed the sequence length of all source and target sentences to be 30, and trained models with 100 million dummy tokens without validation (It takes 920 steps to train each model).

**Base Model**    The base SCoMoE was trained on a single node, hence we only need to consider the ratio for the intra-accelerator communication ($\alpha$). For a fair comparison on translation performance, we set $\alpha$ for SCoMoE-Seq to 0.3, and that for SCoMoE-Feat to 0.5, the capacity factor of Small-Cap-

---

[4] https://github.com/pytorch/fairseq/blob/moe/examples/translation/prepare-wmt14en2fr.sh
[5] https://opus.nlpl.eu/opus-100.php
[6] https://github.com/pytorch/fairseq/blob/moe/fairseq/benchmark/dummy_mt.py

Table 1: Translation performance of the base and large models on the test sets of OPUS-100 and WMT17-En-Fr. SCoMoE-(Seq/Feat)-$\alpha$: the SCoMoE-(Seq/MT) with the intra-accelerator and global communication. Time refers to the average time cost (ms) of all-to-all communication.

| Setting | Model | Multilingual | | | | Bilingual | speedup | Time |
|---------|-------|-----|--------|--------|-------------|-----------|---------|------|
| | | All | En-Any | Any-En | win-rate (%) | | | |
| Base (0.2B) | Transformer | 23.43 | 19.64 | 27.17 | 26.2 | 37.39 | 2.00x | - |
| | Gshard | 24.41 | 20.02 | **28.76** | - | 37.96 | 1.00x | 1.6 |
| | Small-Cap-MoE | 23.02 | 19.08 | 26.92 | 20.3 | 37.27 | 1.10x | 1.33 |
| | SCoMoE-Seq-$\alpha$ | **24.45** | **20.67** | 28.18 | **43.3** | **38.24** | 1.10x | **1.14** |
| | SCoMoE-Feat-$\alpha$ | 23.47 | 20.25 | 26.67 | 27.3 | 37.51 | 1.07x | **1.13** |
| Large (4.1B) | Gshard | 26.78 | 23.11 | 30.44 | - | 40.09 | 1.00x | 27.0 |
| | Small-Cap-MoE | 26.93 | 24.58 | 29.28 | 49.5 | 39.87 | 1.20x | 21.3 |
| | SCoMoE-Seq-$\alpha$ | 27.19 | 25.68 | 28.69 | 69.5 | 39.31 | 1.20x | 20.3 |
| | SCoMoE-Feat-$\alpha$ | 27.99 | 25.13 | 30.85 | 73.9 | 39.79 | **1.32x** | 17.4 |
| | SCoMoE-Seq-$\beta$ | **29.60** | **27.32** | **31.39** | **90.4** | **40.20** | 1.25x | **16.8** |
| | SCoMoE-Feat-$\beta$ | 28.19 | 24.90 | **31.47** | 71.2 | 39.93 | 1.10x | 19.9 |

MoE to 0.8, so that these models can be trained with a comparable training speed. More evaluation on the training speed of SCoMoE can be found in Appendix E

From Table 1, we can observe that both SCoMoE-Seq/Feat and Small-Cap-MoE are faster than Gshard by nearly 10%. But SCoMoE cost less communication time than Small-Cap-MoE. Why do they still have similar training speed? The reason is that SCoMoE brings some extra overhead on computation, which is analyzed in Appendix E. Although SCoMoE-Feat is slightly slower than Small-Cap-MoE, it is better in terms of BLEU on both the WMT17-En-Fr and OPUS-100 dataset. While SCoMoE-Seq is comparable or even better than Gshard on both the WMT17-En-Fr and OPUS-100 dataset, due to the proposed token clustering and differentiable sorting. We further demonstrate the effectiveness of the two methods in Appendix F and G.

**Large Model** The large SCoMoE models were trained on multiple nodes, hence both SCoMoE-Seq-$\alpha$ and -$\beta$ were evaluated. We set $\alpha = 0.2$ for SCoMoE-Seq-$\alpha$ and 0.5 for SCoMoE-Feat-$\alpha$, which have the comparable training speed with SCoMoE-Seq/Feat-$\beta$ with $\beta = 0.5$. We set the capacity factor of Small-Cap-MoE to 1.0, while that of other models was set to 1.25. From Table 1, it is clear that all variants of SCoMoE are faster and consume less communication time than Gshard significantly. Among the 4 variants, SCoMoE-Feat-$\alpha$ is the fastest. It is reasonable since we set $\alpha$ of SCoMoE-Feat-$\alpha$ to 0.5, but that for SCoMoE-Seq-$\alpha$ to only 0.3. We provide the training speed comparison of SCoMoE with more settings of $\alpha$ and $\beta$ in Appendix E. It can also be seen that SCoMoE-Seq-$\alpha$ is faster than SCoMoE-Seq-$\beta$ but takes more communication time. This is because that SCoMoE uses the top-2 routing, where each token is computed in two experts. In comparison, tokens for intra-accelerator communication are computed in a single expert inside the accelerator, which reduces computational time.

We have also found that all variants of SCoMoE achieve better translation performance than both Gshard and its variant Small-Cap-MoE on the OPUS-100 dataset. They also perform comparably to Gshard on the WMT17-En-Fr dataset. Among the 4 variants of SCoMoE, SCoMoE-Seq-$\beta$ is the best in terms of BLEU on both OPUS-100 and WMT17-En-Fr. It is surprising that SCoMoE-Seq-$\beta$ outperforms Gshard by 2.8 BLEU on OPUS-100, but only 0.1 BLEU on WMT17-En-Fr. We conjecture that MoE models are easy to overfit on low-resource languages (Costa-jussà et al., 2022), while our method can mitigate this issue by regularizing the routing of MoE. We have shown the improvements of SCoMoE on different languages in Appendix D to validate this hypothesis. Overall, all variants of SCoMoE are faster than Gshard with comparable or better translation performance.

**The best settings of $\alpha/\beta$ for SCoMoE** From Table 1, we can roughly see that SCoMoE-Seq/Feat-$\beta$ have better translation performance, while SCoMoE-Seq/Feat-$\alpha$ are faster. We further compared the translation performance of them with different settings of $\alpha$ or $\beta$. The results are shown in Figure 3a and 3b. According to Figure 3a, SCoMoE-Seq-$\alpha$ is better with a smaller $\alpha$ in terms of BLEU, which is not surprising since increasing $\alpha$ will reduce the amount of data for global communication. But SCoMoE-MoE-$\beta$ achieves better translation performance with larger $\beta$s. Furthermore,

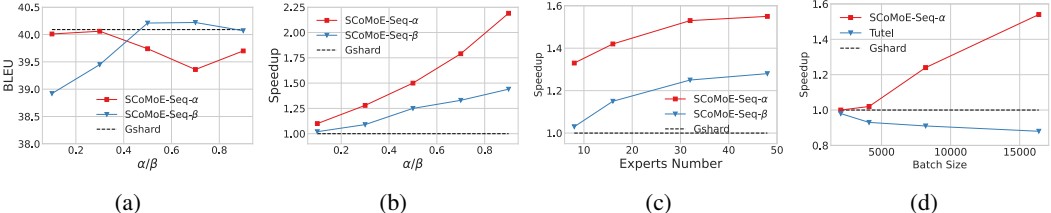

Figure 3: BLEU and Speedup of SCoMoE at different settings. (3a): BLEU of SCoMoE-Seq-$\alpha/\beta$ with different $\alpha/\beta$. (3b): Speedup of SCoMoE-Seq-$\alpha/\beta$ with different $\alpha/\beta$. (3c): Speedup of SCoMoE-Seq-$\alpha/\beta$ with different number of experts. (3d): Speedup of SCoMoE-Seq-$\alpha$ with different batch sizes.

SCoMoE-MoE-$\beta$ is even comparable to Gshard when $\beta = 0.9$, suggesting that only a small proportion (10%) of tokens need to be communicated in global communication and that intra-node communication is sufficient for the majority of tokens. SCoMoE-Seq-$\beta$ with small $\beta$s underperforms Gshard, the reason for which can be found in Table 4 where Gshard also suffers from performance drop with token clustering and reordering.

With regard to the comparison on the training speed (Figure 3b), the speedup of SCoMoE-seq-$\alpha/\beta$ over Gshard increases nearly linearly with the increment of $\alpha/\beta$. The rate of increment for SCoMoE-seq-$\alpha$ is around 1.0, while that for SCoMoE-seq-$\beta$ is 0.5. Therefore, to achieve the best speedup without performance drop in BLEU, we can set $\alpha$ of SCoMoE-seq-$\alpha$ to 0.3 and $\beta$ of SCoMoE-seq-$\beta$ to 0.9, which can lead to a speedup of 1.28x and 1.44x respectively. We can also set $\beta$ of SCoMoE-seq-$\beta$ to 0.5, which achieves the highest BLEU scores and a speedup of 1.25x.

**Scaling in model and batch size** The speedup of SCoMoE is also dependent on the model size and batch size. Figure 3c and 3d compares the training speed of SCoMoE with different number of experts and batch sizes. From Figure 3c, it is clear that the speedup of SCoMoE increases with more experts, but the trend is not so significant when the number of experts increases to 48. It is due to that the communication cost is more expensive with more devices. Similarly, SCoMoE also has a more significant speedup with larger batch sizes, as shown in Figure 3d. In Figure 3d, we also display the training speed of Gshard with the hierarchical all-to-all communication proposed in Tutel (Hwang et al., 2022). However, the hierarchical communication of Tutel even slows down the training. The reason may be that the intra-node bandwidth in our experiments is not fast (PCIe) and that the number of devices is not very large (32).

**Other Experiments and Analyses** In addition to the main results, we conducted further experiments and analyses, which are presented in Appendix, including the analysis on why SCoMoE can improve translation performance (D), extensional evaluation on training and translation speed of SCoMoE (E, J), ablation study (F,G,H), tuning $\alpha, \beta, \gamma$ simultaneously (I) and performance vs. time evaluation (K).

## 5 CONCLUSION

In this paper, we have presented SCoMoE to reduce the all-to-all communication time of MoEs, by encouraging more data to dispatch through the fast intra-accelerator/node communication channel, and less data through the slow inter-node communication channel. By slicing/projecting the data on the sequence/feature dimension, we transform the data into three groups corresponding to three levels of communication. To alleviate the performance drop caused by routing locality, we further propose the token clustering to aggregate related tokens from different devices, and replace the sigmoid gate of the balanced router with the softmax gate through differentiable sorting. Experiments on bilingual and massively multilingual neural machine translation demonstrate that our SCoMoE speeds up Gshard significantly with comparable or even better performance. The empirical evaluation and theoretical analysis on the training speed and communication time cost discloses the speedups achieved by our method under different setting and network bandwidth. The ablation study demonstrates the benefit of the proposed token clustering and differentiable sorting.

## ACKNOWLEDGEMENTS

The present research was supported by the Key Research and Development Program of Yunnan Province (Grant No. 202203AA080004) and the joint research center between GTCOM and Tianjin University. We would like to thank the anonymous reviewers for their insightful comments.

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

## APPENDIX

## A    RELATED WORK ON MASSIVELY MULTILINGUAL MACHINE TRANSLATION

Recent years have witnessed an upsurge of interest in multilingual neural machine translation (MNMT) that trains a single NMT model to translate multiple languages (Johnson et al., 2017; Ha et al., 2016; Aharoni et al., 2019; Arivazhagan et al., 2019). Efforts have been devoted to pushing the limits of MNMT in terms of the number of languages supported by adding language-specific parameters (Wang et al., 2018; 2019; Zhang et al., 2020; Jin & Xiong, 2022; Xu et al., 2021), adapters (Bapna & Firat, 2019; Zhu et al., 2021; Zhang et al., 2021a; Sun & Xiong, 2022), or increasing the depth (Huang et al., 2019) and width Fan et al. (2021) of the model. In addition to these, MoE has become a popular architecture to scale MNMT to massively MNMT (Lepikhin et al., 2020; Kim et al., 2021; Costa-jussà et al., 2022). In this work, we also conduct experiments on massively MNMT to show the effectiveness of SCoMoE.

## B    DIFFERENTIABLE SORTING

The auction algorithm in the balanced router is not differentiable, hence the BASE Layer (Lewis et al., 2021) learns the expert embedding via the sigmoid gate (Srivastava et al., 2015):

$$\boldsymbol{X}_i' = \sigma(\boldsymbol{M}_{ij}) \cdot \text{expert}_j(\boldsymbol{X}_i) + (1 - \sigma(\boldsymbol{M}_{ij})) \cdot \boldsymbol{X}_i \qquad (4)$$

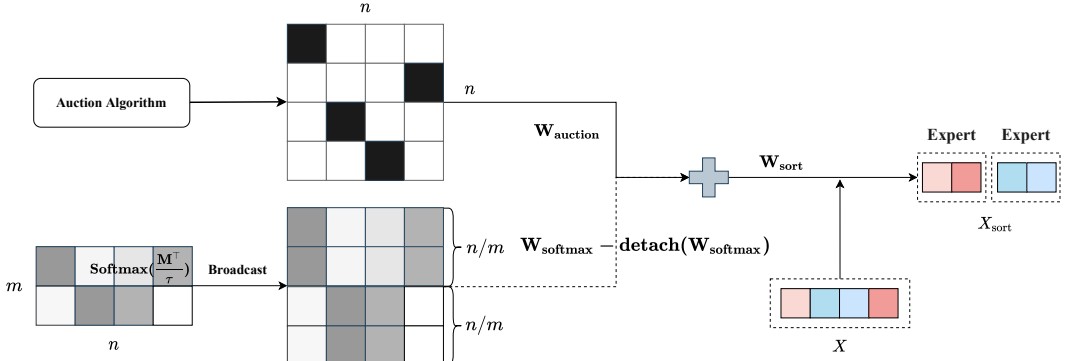

Figure 4: The visualization of differentiable sorting. The sorting projection $\boldsymbol{W}_{\text{sort}}$ from the auction algorithm, and the broadcast probability matrix $\boldsymbol{W}_{\text{softmax}}$ are combined together via the straight-through trick to sort the sequence. The sorted four tokens are divided into two groups, per corresponding to one expert.

where the output of the MoE layer $\boldsymbol{X}_i'$ is the combination of the expert input $\boldsymbol{X}_i$ and expert output $\text{expert}_j(\boldsymbol{X}_i)$ that are integrated with the normalized assignment score $\boldsymbol{M}_{ij}$. The gradients are broadcast through the sigmoid gate to update the parameters of the expert embeddings. But only the embeddings of the activated experts can receive the gradients, even though we have computed the assignment scores between any tokens and experts. With the gradients from the sigmoid gate, we only know how to update the assignment scores of the activated experts, which is not available for other experts. A straightforward solution to this is to replace the sigmoid gate with the softmax gate. However, the softmax operator requires multiple inputs, but only one expert is activated for each token. We notice that the auction algorithm assigns tokens to experts via sorting, which could be implemented as a linear projection with a sparse matrix, where each row and column only contains one non-zero value. Each row of the matrix could be seen as selecting a token from the sequence for an expert. We implement the sorting via linear projection:

$$\boldsymbol{X}_{\text{sort}} = \boldsymbol{W}_{\text{sort}} \cdot \boldsymbol{X} \tag{5}$$

where the sorted sequence $\boldsymbol{X}_{\text{sort}}$ is obtained by the projection $\boldsymbol{W}_{\text{sort}}$ on the input sequence $\boldsymbol{X}$. If the number of sorted tokens is $n$, and the number of experts is $m$, $n/m$ tokens will be assigned to each expert. We visualize the sorting with a $4 \times 4$ sparse projection on a sequence of length 4 in Figure 4. The tokens in the sorted sequence are assigned to two experts. The assigned tokens often have the max or top-$k$ scores with the expert, therefore each row could be seen as the argmax or argtopk on the token-to-expert assignment matrix. The softmax is the approximation to the argmax, if we set the temperature to a small value. Therefore, we share the gradients of the projection matrix with the softmax matrix via the straight-through trick (Bengio et al., 2013):

$$\boldsymbol{W}_{\text{sort}} = \boldsymbol{W}_{\text{auction}} + \boldsymbol{W}_{\text{softmax}} - \text{detach}(\boldsymbol{W}_{\text{softmax}})$$
$$\boldsymbol{W}_{\text{softmax}} = \text{softmax}(\frac{\boldsymbol{M}^\top}{\tau}) \tag{6}$$

$\boldsymbol{W}_{\text{sort}}$ is the combination of the sorting projection $\boldsymbol{W}_{\text{auction}}$ from the auction algorithm, $\boldsymbol{W}_{\text{softmax}}$, which is the normalized assignment matrix, and $\text{detach}(\boldsymbol{W}_{\text{softmax}})$, which has the same value as $\boldsymbol{W}_{\text{softmax}}$ but cannot broadcast the gradients. $\tau$ is the temperature which controls the sharpness of the softmax distribution. The lower the temperature is, the better the softmax approximates the argmax. $\boldsymbol{W}_{\text{sort}}$ is equal to $\boldsymbol{W}_{\text{auction}}$ in the forward computation. $\boldsymbol{W}_{\text{softmax}}$ has the same gradients as $\boldsymbol{W}_{\text{sort}}$ in the backward propagation, although $\boldsymbol{W}_{\text{softmax}}$ does not have influence on the training loss.

As the shape of the probability matrix $\boldsymbol{W}_{\text{softmax}}$ is different from that of $\boldsymbol{W}_{\text{sort}}$, $(m, n)$ vs. $(n, m)$, we cannot add them directly. Each row of $\boldsymbol{W}_{\text{softmax}}$ is a set of probabilities assigning tokens to an expert, which are corresponding to $n/m$ rows of $\boldsymbol{W}_{\text{sort}}$ as we need to assign $n/m$ tokens to each expert according to the same distribution. For the example in Figure 4, both the first and second row of sparse matrix $\boldsymbol{W}_{\text{auction}}$ are corresponding to the first row of the probability matrix $\boldsymbol{W}_{\text{softmax}}$. We reshape $\boldsymbol{W}_{\text{softmax}}$ from $(m, n)$ to $(n, n)$ by repeating each row of the matrix to $n/m$ rows. In Figure 4, we reshape the matrix from (2,4) to (4,4), by repeating the first and second row of the matrix $\boldsymbol{W}_{\text{softmax}}$ to the first two and second two rows of the broadcast matrix.

## C  DETAILS ON EXPERIMENT SETTINGS

Table 2: The statistics of the WMT17-En-Fr and OPUS-100 dataset.

| Datasets | Languages | Train | Validation | Test |
|---|---|---|---|---|
| WMT17-En-Fr | En-Fr | 35,762,532 | 26,854 | 3003 |
| OPUS-100 | 100 | 107,924,846 | $188 \times 1000$ | $188 \times 1000$ |

Table 3: The configuration of base and large models. Enc/Dec-Layer denotes the number of encoder/decoder layers.

| Setting | Enc-Layer | Dec-Layer | $d_{\text{model}}$ | Experts | GPU | Nodes | Parameters (B) |
|---|---|---|---|---|---|---|---|
| Base | 6 | 6 | 512 | 8 | 8 | 1 | 0.2 |
| Large | 12 | 12 | 1024 | 32 | 32 | 4 | 4.1 |

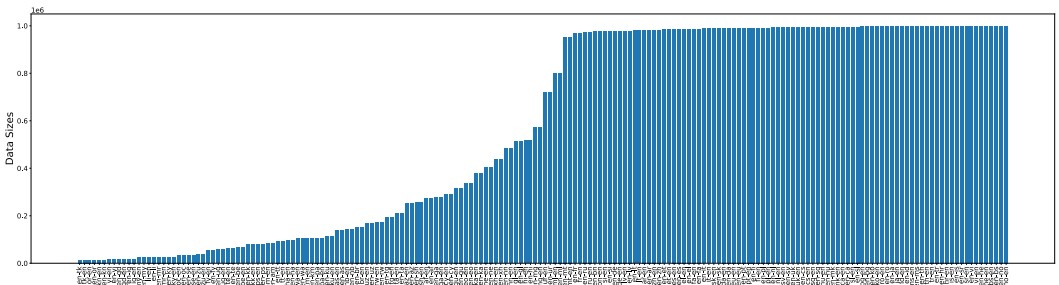

Figure 5: Data distribution of different languages in OPUS-100.

We used the MoE branch of fairseq [7] to implement our MoE models. We prevented the padding tokens from being sent to experts by masking their assignment scores. The capacity factor of the base model was set to 1, while that of the large model was set to 1.25 following the suggestion of (Zoph et al., 2022). The tokens with small assignment scores to an expert were dropped if the expert was out-of-capacity. All models used pre-normalization to stabilize the training. We have found that the MoE models are unstable to train in the large setting, hence we reduced the gradient norm by gradient clipping (i.e., gradient norm was clipped to be no more than 0.5) and gradient accumulation (accumulating the gradients every 4 steps). The gradients of experts were divided by the square root of the number of accelerators, which also reduced the gradient norm and may improve the stabilization of training. We have also found that the MoE model trained in multilingual experiments could not perform well at inference, and that increasing the load-balance loss weight from 0.01 (default value) to 0.1 significantly improves the performance. Therefore, we set the load-balance loss weight to 0.1 in the multilingual experiments, and 0.01 in the bilingual experiments.

The Adam optimizer (Kingma & Ba, 2015a) with learning rate $= 5e^{-4}, \alpha = 0.9, \beta = 0.98$, was used to optimize our models. We employed the inverse-square-root learning schedule (Kingma & Ba, 2015b), with the number of warm-up step being set to 4000. The batch size was set to be 4096 tokens. The dropout rate was 0.3 for bilingual experiments and 0.1 for multilingual experiments. Both the base and large models were trained for 10 epochs. The checkpoint with the best performance on the validation set was selected for testing. In the bilingual experiments, we evaluated the performance on the validation set with BLEU. In the multilingual experiments, the perplexity (ppl) on the validation set was used for faster validation. The BLEU score was computed via sacrebleu (Post, 2018).[8]

---

[7]https://github.com/pytorch/fairseq/tree/moe

[8]The signature of the sacrebleu:BLEU+case.mixed+lang.xx-xx+numrefs.1+smooth.exp+tok.13a+version.1.5.1. xx-xx denotes the language direction, e.g., en-fr

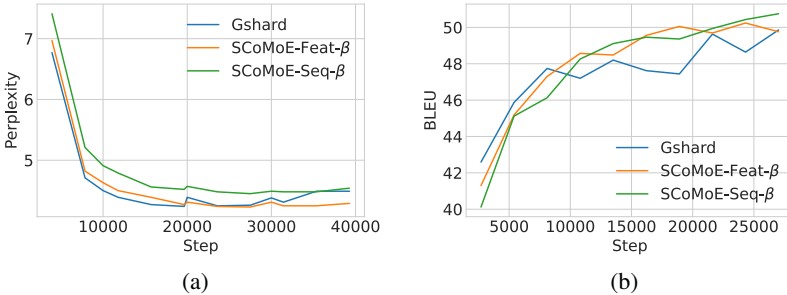

(a)                (b)

Figure 6: Convergence curves of Gshard and SCoMoE on the validation set of OPUS-100 (6a) and WMT17-En-Fr (6b).

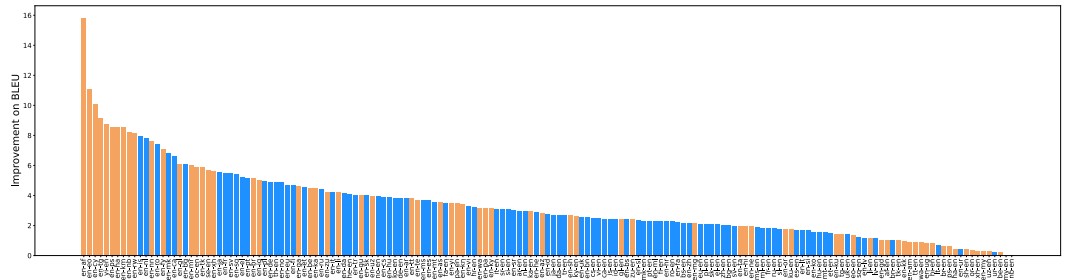

Figure 7: Improvements of SCoMoE-Seq-$\beta$ over Gshard in terms of BLEU on different languages of OPUS-100. Blue/yellow indicate that the language is a rich-/low-resource language.

## D    WHY SCoMoE IS BETTER ON MULTILINGUAL MACHINE TRANSLATION

In Table 1, the improvement of SCoMoE-Seq-$\beta$ on multilingual OPUS-100 is much more significant than that on bilingual WMT17-En-Fr. We plot the convergence curves of Gshard and SCoMoE on validation set of OPUS-100 and WMT17-En-Fr in Figure 6a and 6b respectively. It is clear that the convergence curves of SCoMoE are more smooth than Gshard on both OPUS-100 and WMT17-En-Fr. Gshard obtains its best perplexity on OPUS-100 at the 20kth step, but its perplexity increases as the training continues. While the convergence curve of Gshard on WMT17-En-Fr is also fluctuated, it tends towards getting better. Therefore, Gshard only overfits on OPUS-100. This is reasonable, since WMT17-En-Fr contains much more data than any single language pair in OPUS-100 does. In Figure 5, we illustrate the data distribution of different language pairs in OPUS-100. It is clear that only half of language pairs have about 1 million parallel sentence pairs, while other language pairs have much less data. We also show the improvements of SCoMoE over Gshard on different language pairs of OPUS-100 in Figure 7, where we visualize different language pairs in different colors according to whether the language is low-resource or rich-resource. We define a language pair as a low/rich-resource language pair according to whether the data size of this language pair is larger than the median size (640k) of all language pairs. From Figure 7, we can observe that the large improvements ($>$ 8 BLEU) of SCoMoE are all achieved on the low-resource language pairs. Among these language pairs, en-af only has 274,821 sentence pairs, but gets the greatest improvement of nearly 16 BLEU. All of these suggest that SCoMoE alleviates the overfitting issue of Gshard especially on low-resource language pairs.

## E    EVALUATION ON TRAINING SPEED AND COMMUNICATION TIME

In this section, we compared the training speed and communication time of SCoMoE-Seq and SCoMoE-Feat with different settings of $\alpha$ and $\beta$. We report the training speed in the metric of words-per-second (WPS), which is the number of tokens that the model can process per second, and the time cost of each all-to-all communication in milliseconds.

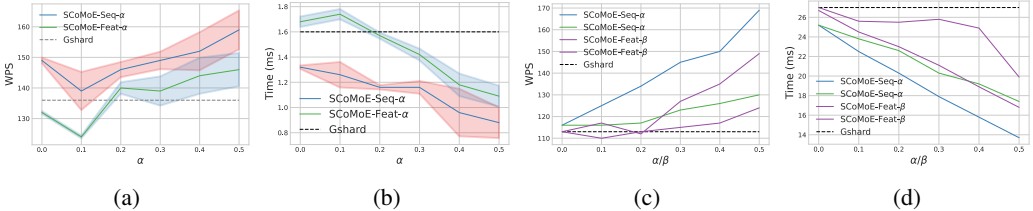

Figure 8: (8a): The speed of base SCoMoE with different $\alpha$. (8b): The communication time of base SCoMoE with different $\alpha$. (8c): The speed of large SCoMoE with different $\alpha/\beta$. (8d) The communication time of large SCoMoE with different $\alpha/\beta$. SCoMoE-Seq/Feat-$\alpha$ denotes the SCoMoE-Seq/Feat with the intra-accelerator and global communication, for which the x-axis is the ratio of the intra-accelerator communication ($\alpha$). SCoMoE-Seq/Feat-$\beta$ denotes the SCoMoE-Seq/Feat with the intra-node and global communication, for which the x-axis is the ratio of the intra-node communication ($\beta$).

**Base Model** We tuned the ratio of the intra-accelerator communication ($\alpha$) from 0 to 0.5, and illustrated the training speed and communication time of SCoMoE at different ratios in Figure 8a and 8b respectively. In spite of training with fixed dummy inputs, the training speed of the base models is still unstable. Hence we trained the same model for three times and show the mean and the standard deviation over the three runs in Figure 8a and 8b.

From Figure 8a, we observe that the two models can be trained faster as the ratio of intra-accelerator communication ($\alpha$) increases, except that the training of both SCoMoE-Seq and SCoMoE-Feat becomes slower when the ratio is 0.1, but with less communication time than when the ratio is 0. This suggests that the structured communication reduces the communication time but may cause additional overhead. The extra overhead is probably from the computation, since computations in each expert are also split into multiple groups corresponding to multiple communication groups, which is slower than performing all computations at the same time. Even if we set the ratio of the intra-accelerator communication to 0, SCoMoE-Seq is still significantly faster than Gshard. The reason is that the communication cost of token clustering is half as that of the top-2 routing. We also notice that the SCoMoE-Seq is always faster than SCoMoE-Feat, and requires less communication time. This is because the token clustering is only applied to the SCoMoE-Seq.

**Large Model** Both SCoMoE-Seq/Feat-$\alpha$ and -$\beta$ can be used for training large models. The differences between them is that SCoMoE-Seq/Feat-$\beta$ accelerate the training speed with the intra-node communication instead of the intra-accelerator communication. We have compared the training speed and communication time of them with different settings of $\alpha/\beta$ in Figure 8c and 8d respectively. Overall, all variants of SCoMoE train faster and require less communication time as $\alpha$ or $\beta$ increases. It is also clear that SCoMoE-Seq/Feat-$\alpha$ is always faster than SCoMoE-Seq/Feat-$\beta$. This is not surprising as the intra-accelerator communication is much faster than the intra-node communication. But we also notice that the training speed and communication time of SCoMoE-Feat-$\beta$ is not significantly better than Gshard until $\beta$ reaches 0.5, while that for SCoMoE-Seq-$\beta$ linearly scales as $\beta$ increases. The reason for this might be that the amount of communication time is related to the shape of communicated tensor. It seems that the communication will be faster if the feature dimension is the power of 2.

**Theoretic Analysis on Communication Time** We have empirically evaluated the training speed and the amount of communication time of SCoMoE by varying $\alpha$ and $\beta$. The bandwidth of communication has a great impact on these empirical results in practice. To complement the empirical evaluation, we further theoretically analyze the amount of time required by the all-to-all communication and estimate speedups that can be achieved by the proposed structured communication method. We estimate the all-to-all communication time as the combination of the intra-node, and inter-node communication time as the time cost for the intra-accelerator communication (memory copy) is negligible in comparison to the other two communication channels. Suppose that the total amount of data for communication is $D$ MB, and the number of accelerators in one node is $m$. Given that, the amount of data for the intra-/inter-node communication is $\frac{D}{m}/\frac{D(m-1)}{m}$. Suppose that the bandwidth of the intra/inter-node communication is $B_1/B_2$ MB/s. The amount of the all-to-all communication

time (seconds) can be estimated as follows:

$$
\begin{aligned}
T_{\text{all-to-all}} &= \frac{D}{mB_1} + \frac{D(m-1)}{mB_2} \\
&= \frac{D(m-1)}{m}\left(\frac{1}{(m-1)B_1} + \frac{1}{B_2}\right)
\end{aligned}
\tag{7}
$$

According to the equation, increasing the bandwidth of the inter-node communication ($B_2$) is more beneficial than increasing the bandwidth of the intra-node communication ($B_1$) as $\frac{1}{m-1} \leq 1$. If we set the ratio of the intra-node communication to $\beta$ ($\alpha = 0, 0 < \beta < 1$), the amount of the all-to-all communication time can be computed as:

$$
\begin{aligned}
T_\beta &= (1-\beta)T_{\text{all-to-all}} + \beta\frac{D}{B_1} \\
&= (1-\beta)\left(\frac{D}{mB_1} + \frac{D(m-1)}{mB_2}\right) + \beta\frac{D}{B_1} \\
&= (1-\beta)\frac{D(m-1)}{m}\left(\frac{1}{B_2} - \frac{1}{B_1}\right) + \frac{D}{B_1}
\end{aligned}
\tag{8}
$$

The absolute value of the derivative of $\beta$ is proportional to $\frac{1}{B_2} - \frac{1}{B_1}$. Either decreasing $B_2$ or increasing $B_1$ would make the SCoMoE more efficient. Therefore, If the bandwidth of the intra-node communication is slow, it is better to set $\beta = 0$. Typically, there are two types of intra-node connections with different bandwidths: PCIE and NVlink/NVSwitch. The NVlink/NVSwitch connection is much faster than the PCIE connection. Furthermore, the network interface controller (NIC) is also connected with accelerators through PCIE. More communications sharing PCIE connections, will make the communication slower. Therefore, we recommend to set $\beta > 0, \alpha = 0$ if fast intra-node connections with NVlink/NVSwitch are available, and $\alpha > 0, \beta = 0$ if slow intra-node connections with PCIE are used. We use the PCIE connections in our experiments, hence the speedup of SCoMoE-$\beta$ should be more significant with the fast NVlink/NVSwitch connections.

## F  EFFECTS OF TOKEN CLUSTERING

Table 4: The translation performance and training speed of Gshard and SCoMoE-Seq with/without token clustering and layer reordering. ✓/ × denotes with/without. Clustering denotes the token clustering. Reordering denotes changing the order of layers according to the architecture in Figure 2

| Model | Clustering | Reordering | Multilingual | Bilingual | Speed |
|---|---|---|---|---|---|
| Gshard | × | × | 24.41 | 37.96 | 1x |
| Gshard | × | ✓ | 22.73 | 38.29 | 1x |
| Gshard | ✓ | ✓ | 22.86 | 37.93 | 1.1x |
| SCoMoE-Seq | × | × | 22.81 | 38.0 | 1.05x |
| SCoMoE-Seq | ✓ | ✓ | **24.45** | **38.24** | 1.1x |

We compared the translation performance and training speed of Gshard and SCoMoE with vs. without token clustering. Results are shown in Table 4. It could be seen that the Gshard with token clustering or layer reordering performs comparably to Gshard without using them on bilingual translation, but is significantly worse than the original GShard on multilingual translation. Results of SCoMoE demonstrate that token clustering can significantly benefit our model on both bilingual and multilingual machine translation in terms of both BLEU and training speed. The reason for that Gshard with token clustering and reordering has the same training speed (1.1x) with SCoMoE are in two folds: 1) Token clustering speeds up the training of both Gshard and SCoMoE, since the communication cost of token clustering is only half as that of Top2Gate routing. 2) SCoMoE brings additional computational overhead and requires a setting of higher $\alpha$ to be faster than Gshard, which has been discussed in Appendix E.

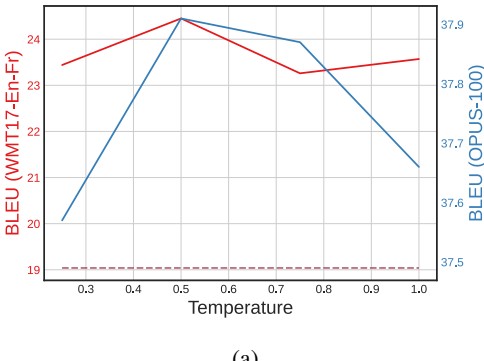
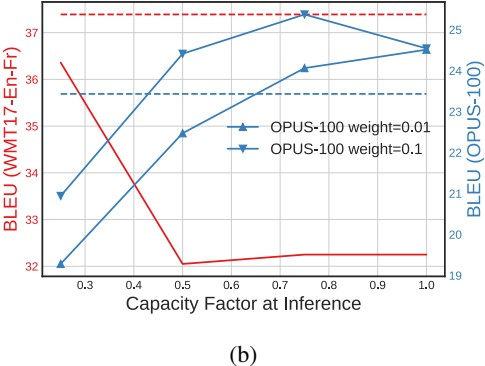

(a)                                                                                      (b)

Figure 9: (a) The translation performance of SCoMoE-Seq with different temperatures on the test set of OPUS-100 and WMT17-En-Fr. The horizontal dashed line denotes the BLEU scores of the SCoMoE-Seq with the sigmoid gate on OPUS-100 and WMT17-En-Fr. (b) The translation performance of Gshard with different eval-capacity-factor. The two horizontal dashed lines denote the BLEU scores of the standard Transformer on OPUS-100 (blue) and WMT17-En-Fr (red). The two dashed lines in Figure (a) are overlapped together.

## G  IMPORTANCE OF DIFFERENTIABLE SORTING

We conducted experiments to investigate how the temperature of softmax affect the performance. We compared the translation performance of SCoMoE-Seq with different temperatures. Results are plotted in Figure 9a, from which we could see that the performance of the model is sensitive to the temperature. We also report the performance of SCoMoE-Seq with the sigmoid gate as the baseline, which is also displayed in Figure 9a (the horizontal dashed line). It could be seen that SCoMoE-Seq with the softmax gate is consistently better than that with the sigmoid gate at different temperatures.

## H  EFFECT OF CAPACITY FACTOR AT INFERENCE

The capacity factor is of importance to the training of MoE models (Yang et al., 2021). In our multilingual NMT experiments, we have found that the models trained on the OPUS-100 dataset performs poorly on the test set, unless with a high capacity factor, substantially higher than that used during training. For convenience, we refer to the capacity factor at inference as eval-capacity-factor. The capacity during training and inference can be computed as follows:

$$\text{capacity}_{\text{train}} = \frac{2 \times \text{number of tokens}}{\text{number of experts}} \times \text{capacity-factor} \tag{9}$$

$$\text{capacity}_{\text{eval}} = \text{number of tokens} \times \text{eval-capacity-factor} \tag{10}$$

We evaluated the trained Gshard model with different eval-capacity-factors and show the performance in Figure 9b. The model for evaluation was trained with the capacity factor of 1.0. The performance of Gshard is even worse than the standard Transformer, when the eval-capacity-factor is smaller than 0.75. Setting eval-capacity-factor=0.75 is equivalent to setting capacity-facor=3.0 if the number of experts is 8, which is very expensive. The costs for communication, computation and memory are all proportional to the capacity factor, hence running Gshard on the OPUS-100 dataset is expensive. We also show the impact of eval-capacity-factor on the translation performance on the WMT17-En-Fr dataset in Figure 9b. It is interesting that the performance of Gshard on WMT17-En-Fr decreases as the eval-capacity-factor increases, which is opposite to the observation on OPUS-100. Increasing the eval-capacity-factor changes the routing of MoE, which may hurt the performance. But why do the the multilingual MoE models require a large capacity? We conjecture that the routing of multilingual MoE is language-specific, i.e., the tokens in related languages are routed to the same experts. In our experiments, each batch contains multiple languages during training, but only one language at inference. Therefore, sentences in the same language are routed to the same experts at inference, which causes these experts to overflow and tokens to be discarded. To validate our hypothesis and alleviate this problem, we increase the load-balance weight

Table 5: The results of SCoMoE with different settings of $\alpha$, $\beta$ and $\gamma$ on the testset of WMT17-En-Fr. SCoMoE-Seq* and SCoMoE-Feat* indicate the models where $\alpha$, $\beta$ and $\gamma$ are all positive.

| Model | $\alpha$ | $\beta$ | $\gamma$ | Speedup | BLEU |
|---|---|---|---|---|---|
| Gshard | - | - | - | 1x | 40.09 |
| SCoMoE-Seq-$\alpha$ | 0.2 | 0 | 0.8 | 1.2x | 39.3 |
| SCoMoE-Seq-$\beta$ | 0 | 0.5 | 0.5 | 1.25x | 40.20 |
| SCoMoE-Seq* | 0.2 | 0.2 | 0.6 | 1.23x | 39.6 |
| SCoMoE-Seq* | 0.2 | 0.5 | 0.3 | **1.41x** | 40.07 |
| SCoMoE-Feat-$\alpha$ | 0.5 | 0 | 0.5 | 1.32x | 39.79 |
| SCoMoE-Feat-$\beta$ | 0 | 0.5 | 0.5 | 1.10x | 39.93 |
| SCoMoE-Feat* | 0.25 | 0.25 | 0.5 | 1.16x | **40.39** |
| SCoMoE-Feat* | 0.25 | 0.5 | 0.25 | 1.26x | **40.50** |

in Gshard from 0.01 to 0.1, and observe significant improvements in Figure 9b. But we still need a high capacity factor for the inference of multilingual MoE models. We leave the issue to our future work.

## I    SETTING $\alpha$, $\beta$ AND $\gamma$ TO BE NON-ZERO SIMULTANEOUSLY

Although there are three communication channels, we have only used two of them in SCoMoE-$\alpha$ ($\alpha = 0$, $\beta > 0$, $\gamma > 0$) and SCoMoE-$\beta$ ($\alpha = 0$, $\beta > 0$, $\gamma > 0$) in our previous experiments, since it is difficult to tune three hyperparameters at the same time. In this section, we investigated whether it is beneficial to set $\alpha$, $\beta$, $\gamma$ to be non-zero at the same time. In Figure 3a, we have observed that $\alpha$ should be set to a small value for a comparable performance with Gshard. Therefore we set $\alpha$ to be 0.2 and 0.25 for SCoMoE-Seq and SCoMoE-Feat respectively. The results are shown in Table 5. Comparing SCoMoE-Seq* with SCoMoE-Seq-$\alpha$ and SCoMoE-Seq-$\beta$, we can find that keeping $\alpha$ and $\beta$ non-zero at the same time could speed up the model training without the performance drop. Furthermore, SCoMoE-Seq* with ($\alpha = 0.25, \beta = 0.5, \gamma = 0.25$) is even better than that with ($\alpha = 0.25, \beta = 0.25, \gamma = 0.5$) in terms of both BLEU and training speed. Therefore, it is better for us to set a small $\alpha$ and a large $\beta$ for SCoMoE-Seq*.

SCoMoE-Feat also benefits from setting $\alpha$, $\beta$ and $\gamma$ to be non-zero at the same time, especially for the improvement on BLEU. It is reasonable because increasing communication channels for SCoMoE means that each token can be computed in more experts. This is similar to increasing heads in multi-head attention.

## J    COMPARISON OF TRANSLATION SPEED

In previous experiments, all reported results on speed are those of training speed. In this section we compared the speed of autoregressive decoding of SCoMoE and analyzed the influence of hyperparameters on translation. For SCoMoE-Seq, the hyperparameters of each communication channel $\alpha, \beta, \gamma$ could be tuned at inference and be different from the values at training. Therefore, we tuned $\alpha$ and $\beta$ for SCoMoE-Seq-$\alpha$ and SCoMoE-Seq-$\beta$ respectively and report BLEU and translation speed in Figure 10a and 10b respectively.

Overall, the speedup of SCoMoE during autoregressive decoding is weaker than that at training. It can be seen that SCoMoE-Seq-$\alpha$ is faster than Gshard consistently at different settings of $\alpha$. SCoMoE-Feat-$\alpha$ also shows a slight speedup over Gshard. However, both SCoMoE-Seq-$\beta$ and SCoMoE-Feat-$\beta$ are slightly slower than Gshard at inference. The reason is that the communication during autoregressive inference is bottlenecked by the frequency rather than the size of communication data at inference. But one exception is that SCoMoE-Seq-$\beta$ with $\beta = 0$ is faster than Gshard due to the token clustering.

With regard to the translation performance, both SCoMoE-Seq-$\alpha$ and -$\beta$ obtain their best performance with $\alpha/\beta = 0$. It is also clear that the performance of them decreases as $\alpha/\beta$ increases.

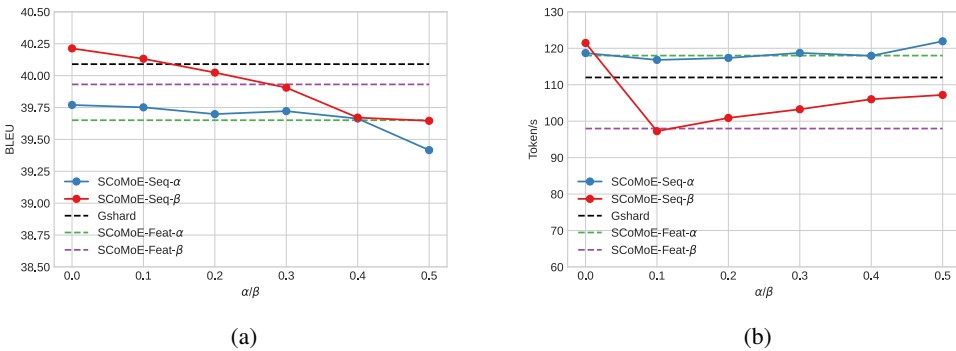

(a)  (b)

Figure 10: Comparison of SCoMoE during autogressive decoding in terms of BLEU (10a) and the decoding speed (10b) on the testset of WMT17-En-Fr. The x-axis ($\alpha/\beta$) represent the hyper-parameters tuning at inference instead of training, which is different from that in Figure 3a and 3b. SCoMoE-Seq/Feat-$\alpha$ and SCoMoE-Seq/Feat-$\beta$ are trained with ($\alpha = 0.5, \beta = 0., \gamma = 0.5$) and ($\alpha = 0., \beta = 0.5, \gamma = 0.5$) respectively.

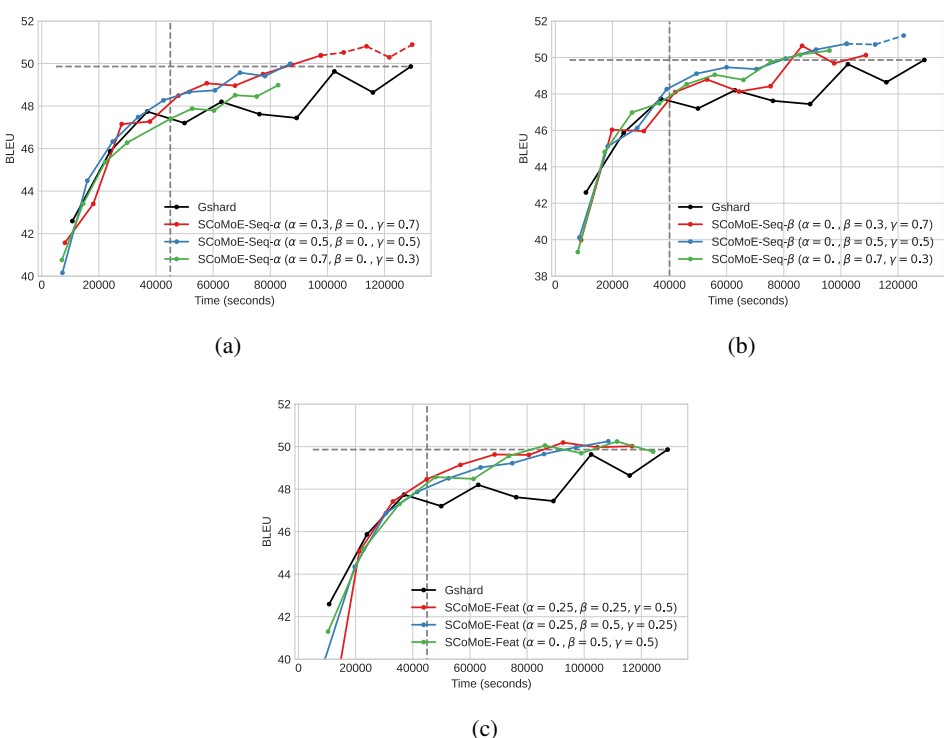

(a)  (b)

(c)

Figure 11: Performance-Time curves of SCoMoE-Seq-$\alpha$ (11a), SCoMoE-Seq-$\beta$ (11b) and SCoMoE-Feat (11c) on validation set of WMT17-En-Fr.

All in all, it is better to set $\alpha/\beta$ to be 0 for SCoMoE-Seq at inference in terms of both translation performance and speedup. We use this setting to report BLEU scores for SCoMoE-Seq in other sections.

## K    PERFORMANCE-TIME CURVES

We have shown the improvements of SCoMoE over Gshard in both training speed and translation performance, but it is still unclear how much time SCoMoE could save compared with Gshard. It is also interesting to see whether SCoMoE could benefit from more training steps. Therefore we compare the performance-time curves of SCoMoE-Seq-$\alpha$, SCoMoE-Seq-$\beta$ and SCoMoE-Feat with Gshard in Figure 11a, Figure 11b and Figure 11c respectively. We can see that almost all SCoMoE variants are superior to Gshard after being trained for 45,000 seconds. And they achieve 50 BLEU at around 80,000 seconds, while Gshard requires 129,000 seconds. Among all variants, SCoMoE-seq with ($\alpha$=0.,$\beta$=0.5,$\gamma$=0.5) and ($\alpha$=0.3,$\beta$=0.,$\gamma$=0.7) seem to be the best. We then trained these two variants with more steps (12 epochs and 15 epochs respectively) so that they were trained with time comparable to that of Gshard (10 epochs). The results of extra steps are illustrated with dashed lines in Figure 11a and Figure 11b. It can be seen that these two variants obtain improvements with more training steps and their final performance is much better than Gshard. Most variants of SCoMoE-Feat converge at around 80,000 steps, and only see little improvements with more training steps.

