# OpenReview forum: "SCoMoE: Efficient Mixtures of Experts with Structured Communication"
_ICLR.cc/2023/Conference — ICLR 2023 poster_

### Official Review · Reviewer_UrrU · 2022-10-21

**Confidence:** 4
**Correctness:** 4
**Technical Novelty And Significance:** 4
**Empirical Novelty And Significance:** 3
**Recommendation:** 8

**Clarity, Quality, Novelty And Reproducibility:**

The paper is clear, and the work is certainly novel and potentially impactful.


**Strength And Weaknesses:**

I have a couple of questions or comments.

First, I think the work would benefit from indeed having cases where both \alpha, \beta and \gamma are *simultaneously* non-zero. Most of the paper describes this case, but I think it's not covered in any of the experiments.

Second, is the speed-up multiplier computed with respect to the total step time? or just the all-to-all communication? Obviously, we care the most about the former. MoE models only have a few sparse layers in general. Thus, saving a 10% wrt to a total of 5% ends up being a very small saving to justify severe engineering extra complexity.

Third (and related to second), if a model saves X% time with respect to another, a nice way to get a sense of how they compare is to run both for the same amount of total time, and look at the final performances. In other words, figure out if the cheap model can re-use the time to improve performance too (or if, unfortunately, the cheap model also has a lower ceiling). A truly informative plot in these cases is performance vs total training time. I'd love to see SCoMoE vs gShard in that type of plot to see how meaningful the gains are.

**Summary Of The Paper:**

The paper proposes a new MoE architecture (and its implementation) that is faster than traditional MoE models. MoE's usually involve sending tokens from one device (where the token is routed) to others (where the selected experts are hosted). This is usually very expensive and one of the main reasons why MoEs are not more popular yet (otherwise the gap in performance vs time wrt dense models would be even larger). This paper proposes a way to partially overcome this issue.

The main idea is to send more tokens to nearby devices. Of course, one needs to be able to choose the right tokens as otherwise the actual extra capacity of having many experts will fade away. There are three types of device communication involved: intra-accelerator, intra-node, and inter-node (from cheaper to more expensive). Assume that every accelerator device contains only one expert. The proposed routing algorithm splits tokens into three groups, those that will be assigned to experts requiring intra-accelerator, intra-node, and inter-node communication, respectively. Note that sending tokens only within a node implies that not all experts will be available to that token (only those in their node!) -- and this will vary depending on where the tokens happen to be. So, if we enforce sending more tokens locally, as in principle those tokens come from a random subset of images, the quality of routing and expert specialization may suffer. The paper acknowledges and confirms this intuition.

The paper proposes two ways to select which tokens are provided with each type of "communication-based" routing. The first one ("sequence-based") aggregates the standard routing weights across experts groups. Depending on the local device for the token, there's experts that require intra-accelerator, intra-node, and inter-node communication, respectively. The router first looks at the sum of weights per token across all the intra-accelerator experts. It ranks the tokens based on this sum, and takes the top-K (where K is a hparam here, a % of the local batch) and applies routing and dispatching to these tokens only, across the intra-accelerator experts. Once those tokens and experts are removed, the same idea is applied to the intra-node experts, and so on. To account for "removed" experts, the missing logits are replaced with -inf before normalizing.

The second routing type, feature-based, projects each token into three representations of different dimensions (corresponding to intra-accelerator, intra-node, and inter-node) summing up to the original dimension. Every expert is also "split" into three smaller subexperts, one that can handle tokens from each of the three dimensions. It's not fully clear to me how the "partitioned tokens" are then routed. Are there three routers (one per communication dimension), and the same logic as for "sequence-based" is applied? If so, different "parts" of a token may end up --not only on different experts-- but on different communication levels?

In order to maximize the chances of having good local routing, the paper proposes to pre-cluster tokens, dispatch similar tokens to the same devices, and then apply the ideas described above. Accordingly, the paper must change the attention-mlp structure to do token clustering only once (so that tokens from the same image can end up in different devices). Figure 2 (left) shows the proposed way to do this. I think this is a cool contribution.

The paper contains a number of experiments for language models where performance and speed are compared between SCoMoE and gShard (a well-known sparse model for language). My takeaway was that SCoMoE can match gShard performance while saving 20-30% of the *overall* time (or is it just all-to-all communication time?), which is quite a nice achievement! They also study what fraction of tokens should be assigned to each of the three groups depending on the routing method.

**Summary Of The Review:**

The paper proposes a new MoE architecture that tries to send most tokens to nearby devices/experts, and applies a couple of tricks to improve the required routing. First, tokens are clustered so that each device gets similar tokens. To make this possible, attention and MLPs are re-arranged so that attention layers aren't applied once MoEs start being used. Two routing techniques are presented, both of which try to identify the tokens that will be better served locally.

It's a nice paper that I really enjoyed. There's a couple of questions regarding the experimental setup and the results whose answer could make the case and my score for the paper stronger.

----------------------------------------------------------------------------------------------
After reading the authors' reply, I've decided to increase my score from 6 to 8.

---

> ### Author Response · Authors · 2022-11-18
> **Response to your questions and suggestions**
>
> Response to Question:
>
> 1. **Are there three routers for each communication channel**?  Yes, it can be said that both SCoMoE-Feat and SCoMoE-Seq use different routers for different communication channels. The difference between different routers is that they target at different experts and they use different expert embeddings to estimate the token-expert assignment scores. If the original expert embedding matrix is $R^{e\times d}$ ($e$ is the number of experts, $d$ is the size of feature dimension), the matrix shapes for the three routers in SCoMoE-Seq are $R^{e_1\times d},R^{e_2\times d},R^{e_3\times d}$ respectively ($e_1+e_2+e_3=e$), while those for SCoMoE-Feat are ${R^{e_1\times d_1},R^{e_2\times d_2},R^{e_3\times d_3}}$  ($d_1+d_2+d_3=d$).
> 2. **Are different parts of a token in SCoMoE-Feat communicated in different levels**? Yes, it is the same as how different groups of tokens are communicated through different channels in SCoMoE-Seq.
>
> Response to weakness
>
> 1. **having cases where both $\alpha$, $\beta$ and $\gamma$ are *simultaneously* non-zero**: Thanks for great your suggestion. We have added a new section with experiment results for that in Appendix H.
> 2. **Is the speed-up multiplier computed with respect to the total step time**? Yes, the speedup reported in this paper are all measured according to the step time rather than the all-to-all time.
> 3. **Missing performance-time curves**: Thanks for your great suggestion. We have added a new section in Appendix J to discuss this.

---

> > ### Comment · Reviewer_UrrU · 2022-11-21
> > **Thanks**
> >
> > Thanks for your reply and updated version of the paper, where you address my questions and comments.
> >
> > Accordingly, I've decided to increase my score from 6 to 8.

---

> > > ### Author Response · Authors · 2022-11-27
> > > **Thank you very much for your valuable comments**
> > >
> > > We are very grateful to you for your great suggestions. Your comments are very important and valuable for us to improve this work!

---

### Official Review · Reviewer_5Gem · 2022-10-23

**Confidence:** 4
**Correctness:** 3
**Technical Novelty And Significance:** 3
**Empirical Novelty And Significance:** 3
**Recommendation:** 6

**Clarity, Quality, Novelty And Reproducibility:**

Some technique presentation is unclear; what is N referring to in Figure 2? I am also a little confused here, is the left side the whole model or a building block of the complete model.

As the author discussed in Section 2, there are some more advanced approaches attempting to optimize the all-to-all communication, there is a lack of corresponding comparison. I think it reasonable to compare the proposed method with some of them, e.g., deepspeed MoE.

There is a lack of code release for reproducibility or a plan to do so.

**Strength And Weaknesses:**

Strength:

+ The idea of using feature project and token clustering is a reasonable relaxation of the original all-to-all communication in the MoE parallel training paradigms.

+ The experiments suggest the good statistical efficiency of the proposed method (in terms of no drop in model quality).


Weakness

+ Some technique presentation is unclear and hard to track.

+ Lack of comparison with some more advanced state-of-the-arts.

**Summary Of The Paper:**

This paper proposes SCoMoE in order to reduce the all-to-all communication in mixture-of-experts model training. The core technique design is based on projecting the feature dimension (SCoMoE-Feat) into low-dimensional representations and using token clustering to compensate for the potential performance drop.

**Summary Of The Review:**

I think the idea in SCoMoE is interesting and suggests good statistical efficiency; however, the presentation of the paper should be improved and some comparison with some more advanced state-of-the-arts should be included to show the hardware efficiency.

---

> ### Author Response · Authors · 2022-11-18
> **Response to your questions and suggestions**
>
> 1. **what is N referring to in Figure 2**? There are $2N$ layers (Attention/FFN/MoE) in SCoMoE and Gshard: $N$ attention layers,  $\frac{N}{2}$ FNN layers and $\frac{N}{2}$ MoE layers.
> 1. **Is the left side the whole model or a building block of the complete model**? The left side of Figure 2 is the whole model structure, while the right side is a building block （SCoMoE layer).
> 1. **Some technique presentation is unclear**. Thanks for your suggestion, we have made the above details more clear in the capation of Figure 2.
> 1. **Lack of comparison with some more advanced state-of-the-arts**: We have indeed compared SCoMoE with a more advanced state-of-the-art, the hierarchical all2all communication in deepspeed MoE (Tutel). The comparison results are shown in Figure 3 (d). Unfortunately, Tutel wasn't able to accelerate the training in our experiments, which may be due to two facts: (1) that the intra-node bandwidth in our experiments is not fast (PCIe) and (2) that the number of devices is not very large (32).
> 1. **There is a lack of code release for reproducibility or a plan to do so.** We have submitted the code and documentation in the new version.

---

> > ### Comment · Reviewer_5Gem · 2022-12-02
> > **Thank you for the feedback!**
> >
> > Dear authors,
> >
> > Thanks for your informative feedback, I think most of my concerns have been resolved, although the baseline could be further tuned given more time. I have raised my score based on the feedback.
> >
> > Best wishes,
> > Reviewer 5Gem

---

### Official Review · Reviewer_2udu · 2022-10-25

**Confidence:** 3
**Correctness:** 3
**Technical Novelty And Significance:** 3
**Empirical Novelty And Significance:** 2
**Recommendation:** 6

**Clarity, Quality, Novelty And Reproducibility:**

The paper is well written, yet as mentioned in the Weaknesses section, my humble opinion is that at least more ablation studies are needed to make the paper more convincing.

Novelty: the motivation is not entirely novel, e.g. compared to TA-MOE: https://openreview.net/pdf?id=FRDiimH26Tr that is similar in the motivation and some other perspectives.

No code is provided so it’s hard to judge the reproducibility. Given the hardware-related nature of the work, and the added methods, I personally doubt the implementation is not complicated, and easy to reproduce.

In addition, by the end of section 2, claiming the two methods can be “easily” combined is probably too strong, esp. without certain evidence.

The chosen track could also be more “ML Systems” rather than NLP applications.


**Strength And Weaknesses:**

Strength: Having a good motivation and direction in ML since many of us work on higher abstraction levels, and will benefit from it in saving time and resources.


Some Weaknesses
1/ The main motivation is not enough without the help from token clustering and the softmax gating + differential sorting in place of sigmoid routing. As a result, it would be helpful to study how each of those 3 contributes to the main results.

2/ Experiments show improvements, but directly compared to Gshard, it seems not so much given how far the solution goes with a combination of methods.

The all-to-all time is very helpful. However, compared to Gshard in terms of practicality of being simplistic, what is the overhead of this solution given the hierarchy and added interfaces? Likewise, it would be also helpful to see the training time, inference time which a typical user would also like to see.


3/ The paper mentions the balancing to some good extent, yet there is no space of analysis to this. Probably it needs a section for regularization of balancing, for clustering might not work perfectly in terms of balancing, and so many tokens must be dropped as some expert reaches its max capacity (assuming the balancing regime is similar to Gshard – please correct if I am wrong).


**Summary Of The Paper:**

This paper has the main motivation to improve the “sharded” distributed MoE by improving its all-to-all communication by not treating them agnostically and equally, but instead characterizing them hierarchically based on their locality relationships, namely accelerator, node and global levels. The motivated application is bi-/multi-lingual machine translation. Experiments show better performance than the baseline Gshard.

**Summary Of The Review:**

The paper has good motivation and direction, as well as their implementation. However, I think it needs improvement to be well convincing.

---

> ### Author Response · Authors · 2022-11-18
> **Response to your questions and suggestions**
>
> Response to the concerns about Clarity, Quality, Novelty And Reproducibility:
>
> 1. **Comparison to TA-MOE**: We did not compare our model with TA-MOE, since it was released in September just before the deadline of ICLR 2023. It is true that both TA-MoE and SCoMoE are based on similar motivations. However, SCoMOE is significantly different from TA-MoE in the following aspects:
>
>     (1) SCoMoE structures the all2all communication into three channels (intra-accelerator, intra-node, inter-node), while TA-MoE penalizes the communication with low bandwidth through an auxiliary loss. Furthermore, the topk-routing in TA-MoE allows the all-to-all communication among different GPUs to have different message size. By contrast, the implementation of (SCoMoE, fariseq, Deepspeed and Tutel) forces different GPUs to communicate with the same message size. In view of this, the implementation of TA-MoE on the topk-routing is confusing for us, which is not available currently (https://github.com/Chen-Chang/TA-MoE).
>
>     (2) SCoMoE transforms the data into multiple groups for different communication channels not only on the sequence dimension but also on the feature dimension, while TA-MoE only considers the sequence dimension.
>
>     (3) Local Routing where more tokens are routed to local devices tends to influence the model perfomance. We propose the token routing to alleviate that, which allows SCoMoE to have the comparable performance with  Gshard even when 90% of tokens are communicated through the intra-node channel.
>
> 2. **About the reproducibility**: We have submitted our code and documentation together with the new version of the paper.
>
> 3. **About the combination of Tutel and SCoMoE**: They can be easily combined because they are implemented at the different levels. The implementation of Tutel is an alternative to the pytorch's all2all communication API,  which can be directly used by SCoMoE. We did not show the results of combination because Tutel does not accelerate the training in our experiments.
>
> Response to the Weakness:
>
> 1. **How each of those 3 contributes to the main results**: We have provided the ablation study for that in Appendix (E, F).
> 2. **Overhead of SCoMoE given the hierarchy and added interfaces**: The main overhead of SCoMoE is that it requires more frequent communication than Gshard, because of the extra intra-accelerator/node communication. This is the reason why the speedup of SCoMoE is weaker with a smaller batch size, as shown in Figure 3 (d), where the speed is bottlenecked by the communication frequency rather than the size of communication data.
> 3. **About the training and inference time**: The speedup reported in our paper are actually all about the training time rather than the all2all communication time.  We have added a new section to show the speedup on the inference time in Appendix I in the new version.
> 4. **Balanced routing in SCoMoE**: For token clustering, we use the balance router of Base Layer [1] rather than that of Gshard, which is totally balanced. While for the routing in other layers, we use the top2gate which is actually unbalanced and results in some tokens dropped. We have analyzed this problem in Appendix G and found that increasing the load-balance loss weight could alleviate this problem.
>
> [1] Lewis, M., Bhosale, S., Dettmers, T., Goyal, N., & Zettlemoyer, L. (2021). BASE Layers: Simplifying Training of Large, Sparse Models. *ICML*.

---

> > ### Comment · Reviewer_2udu · 2022-11-20
> > **Thank you for the rebuttal.**
> >
> > I personally thank you for addressing each of my questions/suggestions. And also thanks for the code, which should be beneficial for the community, although during the short amount of time, I haven't been able to evaluate the code contribution (probably because it's mixed with `fairseq`, which is by no means a bad thing).
> >
> > It's exciting to see more MLSys works like this one, which I think at least the ML community benefits from it. Although I personally prefer simplistic solutions for their practicality and scalability (as I subjectively think MLSys works should be), I appreciate this work, as well as the extended change/additions the author(s) put into this rebuttal. Compare to other works in my batch, I would increase this work to 7 from 6 but there's no such score. So I would maintain my original rating, in favor of seeing this paper appear at the conference.

---

> > > ### Author Response · Authors · 2022-11-27
> > > **Thank you very much for your helpful feedback**
> > >
> > > We are very grateful to you for your great suggestions.  Your comments are very important and valuable for us to improve this work!

---

### Official Review · Reviewer_NyCC · 2022-10-31

**Confidence:** 5
**Correctness:** 3
**Technical Novelty And Significance:** 2
**Empirical Novelty And Significance:** 3
**Recommendation:** 6

**Clarity, Quality, Novelty And Reproducibility:**

The proposed algorithm is described with crystal clarity.  The text, the experimental result tables, and the figure illustrations are all well prepared.

The novelty of this paper comes from taking practical device constraints into the neural architecture design. The authors first identified the bottleneck from an existing architecture, found the root cause, and proposed a novel working solution that addressed the bottleneck.

In addition, I want to call out that the content in the appendix is quite informative as well.


**Strength And Weaknesses:**

Strengths:
1) The authors took practical device constraints into consideration when designing the neural network architecture, and reported real benchmarks instead of just bio-O analyses.

2) As the emergence of ultra-expensive large language models, how to enable affordable large models using more efficient algorithms becomes an important research area.

3) The authors demonstrated the effectiveness of SCoMoE with well designed experiments and evaluations. Real world metrics like BLEU scores and step time are reported. The paper also reported the analysis of capacity factor during inference, which is a critical hyper-parameter for MoE performance.

4) I want to point out that the reported speedup against GShard might seem small (less than 1.5x), however, it's due to the relative smaller number of experts used in the experiments (I guess bounded by compute budget). The speedup achieved by SCoMOE would become more significant when the number of experts is even larger. In the original GShard paper, thousands of experts were used where the communication cost would dominate the total compute cost.


This paper can be even better if the author can address the following concerns:

1) What's the efficiency improvement of this algorithm during autoregressive decoding, where the model decode one token after another sequentially?

2) It's better to estimate how large a dense model needs to be in order to achieve similar translation quality as SCoMoE. This would demonstrate the cost saving from SCoMoE comparing to a dense model with similar quality.

3) Need to discuss the pros and cons of this proposed algorithm comparing to related routing algorithms also aiming to reduce communication costs, such as:
Beyond Distillation: Task-level Mixture-of-Experts for Efficient Inference (https://arxiv.org/pdf/2110.03742.pdf)
Mixture-of-Experts with Expert Choice Routing (https://arxiv.org/pdf/2202.09368.pdf)

**Summary Of The Paper:**

The paper proposed a novel routing algorithm for mixture of experts that took the network bandwidths among the accelerators into consideration, which allows efficient communication for token shuffling especially on the GPUs.


**Summary Of The Review:**

Novel routing algorithm for mixture of experts whose effectiveness were demonstrated by solid experimental results.

---

> ### Author Response · Authors · 2022-11-18
> **Response to your questions and suggestions**
>
> 1. **What's the efficiency improvement of this algorithm during autoregressive decoding**? Thanks for your great suggestion. We have added a new section in Appendix I in the new version to discuss this.
>
> 2. **Compare SCoMoE with dense model**: Thanks for your advice. It is difficult to know how large a dense model should be to achieve the same accuracy as a sparse model. But we could compare the translation performance of  a dense and sparse model with a comparable traning speed, which is shown in Table 1 below. We can see that the dense model that has almost 5 times of shared parameteres as the sparse models, is better than Gshard in terms of both training speed and BLEU. We set $\alpha$, $\beta$ and $\gamma$ to be 0, 0.9 and 0.1 for SCoMoE-Seq respectively, which enables a comparable training speed to the dense model, but a worse translation performance. It seems that dense models are better than the sparse models in current experiments. However,  the dense model in these experiments is trained only with data parallelism. If we want to train a larger dense model, we have to resort to the model parallelism like tensor parallelism, which also requires expensive communication. In this case, the conclusion may be different.
>
> 3. **Comparisons against the two previous works**:
>
>     (1) Comparison with "Beyond Distillation: Task-level Mixture-of-Experts for Efficient Inference":
>
>     - Pros. "Task-level Mixture-of-Experts" extracts a submodel from a multilingual MoE model for each translation direction at inference. However, the extracted submodel could only translate in one direciton. In addition to this, their method only accerlerates the inference speed and only works for multilingual/multi-task MoE. In contrast, SCoMoE could speed up both the training and inference and is not limited to multilingual MoE.
>
>     - Cons. The submodels extracted by "Task-level Mixture-of-Experts" can  be deployed in one accelerator, hence are efficient at inference. By contrast, the speedup of SCoMoE at inference is not as strong as that at training.
>
>     (2) Comparison with "Mixture-of-Experts with Expert Choice Routing": "Expert Choice Routing" selects the top-k tokens for each expert instead of selecting the top-k experts for each token. The benefit of such a change is the better load balance. However, although it seems to converage faster than Gshard, it does not actually save the communication time, since the capacity of each expert in "Expert Choice Routing" is the same as that in Gshard. Although SCoMoE and "Expert Choice Routing" aim to solve different problems, both of them allow different tokens to allocate different amount of computation. In fact, the token slicing in SCoMoE-Seq also selects top-k tokens for experts/devices.
>
>     (3) We have added the above comparisons about them in the section of related work in the uploaded new version.
>
> Table 1. Speed and translation quality comparison of a dense model to two sparse models.
>
>
> | Model      | Shared Params (B) | Expert Params (B) | Speedup   | BLEU (WMT) |
> | ---------- | ----------------- | ----------------- | --------- | ---------- |
> | Gshard     | 0.33              | 3.2               | 1x        | 40.09      |
> | Dense      | 1.54              | 0                 | **1.41x** | **40.78**  |
> | SCoMoE-Seq | 0.33              | 3.2               | **1.41x** | 40.07      |

---

### Author Response · Authors · 2022-11-18
**Response to all reviewers**

Many thanks for your insightful comments and suggestions. We have submmited a new version of the paper together with the code and documentation for our implementation. The main revisions in the new version are listed as follows:

1. A new section (Appendix H) that reports the performance of SCoMoE with $\alpha,\beta$ and $\gamma$ being non-zero simultaneously. Note that we set either $\alpha$  or $\beta$ to be positive in the preivous version, which means that the data are only transformed into 2 rather than 3 groups for different communication channels.
2. A new section (Appendix I) that discusses the speedup of SCoMoE during autoregressive decoding. Note that in the previous version, all reported speedups are measured according to the  whole  training time rather than the all2all communication time.
3. A new section (Appendix J) showing the performance-time curves of SCoMoE.

---

### Decision · Program_Chairs · 2023-01-20

**Decision:**

Accept: poster

**Justification For Why Not Higher Score:**

Limited evaluation of the proposed approach.

**Justification For Why Not Lower Score:**

Novel approach to speedup MoEs and strong results over Gshard.

**Metareview: Summary, Strengths And Weaknesses:**

All reviewers find the proposed approach for training MoEs novel and interesting. The method compares well agains Gshard.  However reviewers also note some limitations in missing comparison to more recent works such as DeepSpeed. Further comparisons are limited to translation task. Despite these limitations, all reviewers felt favorably about the proposed approach and I am happy to suggest acceptance. I encourage authors to include more comparisons in the final version.

Pros -
1. Novel approach to improve MoE efficiency.
2. Good performance in comparison to GShard.

Cons -
1. Limited comparisons to more baselines such as DeepSpeed.
2. Limited evaluation on single translation task.

**Note From Pc:**

if the above contains the word "oral" or "spotlight" please see: "oral" presentation means -> notable-top-5% and "spotlight" means -> notable-top-25%. As stated in our emails, we are disassociating presentation type from AC recommendations

**Summary Of Ac-Reviewer Meeting:**

Reviewers are mainly worried about missing comparisons to other methods such as DeepSpeed, but given the short period for discussion reviewers agreed that it is too much to expect results.